# Microbial Symphony: Navigating the Intricacies of the Human Oral Microbiome and Its Impact on Health

**DOI:** 10.3390/microorganisms12030571

**Published:** 2024-03-13

**Authors:** Rahul Bhandary, Geethu Venugopalan, Amitha Ramesh, Guilia Margherita Tartaglia, Ishita Singhal, Shahnawaz Khijmatgar

**Affiliations:** 1Department of Periodontology, AB Shetty Memorial Institute of Dental Sciences, NITTE (Deemed to be University), Mangalore 575018, India; drrahulbhandary@nitte.edu.in (R.B.); drgeethuv@gmail.com (G.V.); dramitharamesh@nitte.edu.in (A.R.); 2School of Medicine, European University of Madrid, 28670 Madrid, Spain; giugitartaglia@gmail.com; 3Department of Biomedical, Surgical and Dental Sciences, University of Milan, 20129 Milan, Italy; ishita.singhal@unimi.it; 4Fondazione IRCCS Ca’ Granda Ospedale Maggiore Policlinico, 20122 Milan, Italy

**Keywords:** oral microbiome, microbial symphony, human health, microbial colonization, detection methodologies, disease progression

## Abstract

This comprehensive review delves into the forefront of research on the human oral microbiome, exploring recent advancements that span microbial colonization, state-of-the-art detection methodologies, and the complex interplay involved in disease progression. Through an exhaustive analysis of the contemporary literature, we illuminate the dynamic orchestration of microbial communities within the oral cavity, underscoring their pivotal role in health and disease. Cutting-edge detection techniques, including metagenomics and high-throughput sequencing, are discussed regarding their transformative impact on understanding the intricacies of oral microbial ecosystems. As we stand on the cusp of a new decade, this review anticipates a paradigm shift in the field, emphasizing the potential for rapid identification and targeted management of detrimental oral microorganisms. Insights gained from this exploration not only contribute to our fundamental understanding of the oral microbiome but also hold promise for the development of innovative therapeutic strategies to maintain oral health. This article aims to serve as a valuable resource for researchers, clinicians, and public health professionals engaged in unraveling the mysteries of the microbial symphony within the human oral cavity.

## 1. Introduction

The human body, in its intricate complexity, serves as a geographical habitat for various microbial communities, with the potential for the transmission and dissemination of their genetic material across time. The human brain and neurological system, despite their inherent capacity for perceiving and regulating bodily functions, exist within a biological realm known as the body space that serves as a dwelling place for a diverse array of microorganisms, each with its unique genomes [1]. The oral cavity alone harbors a highly intricate, dynamic, and diverse assemblage of microorganisms, surpassing the number of human cells in many instances. The assemblage, referred to as the ‘oral microbiome’, primarily consists of bacteria, archaebacteria (Archaea), viruses, fungi, and protozoa [1,2].

In healthy oral cavities, a conserved community of microorganisms can be observed at the genus level. The unique microenvironment within the oral cavity, characterized by a consistently maintained pH range of 6.5–7.0 in saliva, optimal moisture levels, and an average temperature of 37 °C, fosters the propitious milieu essential for the proliferation of microorganisms [3]. Therefore, a diverse array of microorganisms demonstrates a widespread distribution within the oral cavity. Nevertheless, it is important to note that while there are shared characteristics among individuals, the diversity of these microorganisms is specific to each individual and varies depending on the specific location within the oral cavity [3,4]. Conversely, oral structures tend to facilitate the accumulation of substantial microbial communities, leading to the development of intricate biofilms commonly referred to as plaque [4].

The intricate process of oral microbiome transmission between mother and child begins during pregnancy and continues postnatally. Maternal factors, including the mode of delivery, maternal health, and feeding practices, significantly influence the composition and development of the neonatal oral microbiome. Swallowing of amniotic fluid and early exposure to bacteria from the mother’s oral and placental microbiomes shape the initial colonization of the fetal gut, subsequently influencing the establishment of the oral microbiome. Vertical transmission is evident through correlations between the maternal and neonatal oral microbiomes, emphasizing a direct link. Notably, this transmission is impacted by variables such as maternal exposure to disinfectants and antibiotics during delivery, the type of delivery, maternal overweight status, and gestational diabetes. Although evidence is robust for certain factors, further exploration is required, particularly concerning maternal diet, as well as qualitative assessments of the maternal and neonatal oral microbiome dynamics [5].

Although it is commonly recognized and accepted that microorganisms exist everywhere, microbiologists face significant challenges when capturing the entire microbial community on Petri dishes. This is partly because it is difficult to replicate the natural settings in which these microbes thrive [6,7]. However, due to the remarkable progress in sequencing technology and the emergence of next-generation sequencing (NGS) platforms, the exploration of previously elusive microflorae has now become possible. Considering, the new insights made available by these new molecular and sequencing technologies, a comprehensive database known as the Human Oral Microbiome Database (HOMD) was developed to serve as a repository for essential information on both cultivable and non-cultivable oral microbial isolates [7,8,9]. The updated version of the expanded HOMD (eHOMD) aims to provide extensive knowledge regarding the heterogeneous bacterial populations residing in different oral cavity regions such as the esophagus, pharynx, paranasal air sinuses, and nasal passages [8,9]. These invisible inhabitants have been recognized for their significant interactions with the host cellular entities and demonstrate a direct impact on human physiological processes, metabolic activities, and immune reactions. The significance of altered bacterial communities in the oral cavity is further pronounced when considering the presence of highly perilous health disorders such as obesity, auto-immune diseases, diabetes mellitus, neoplasms, bacteremia, endocarditis, acquired immunodeficiency syndrome (AIDS), etc. [10,11,12,13].

Differentiating between commensal microorganisms and pathogenic ones is a crucial undertaking within the realm of microbiology. The mere identification of pathogenic microorganisms does not invariably culminate in the manifestation of disease. The primary objective revolves around establishing a correlation between the prompt identification of pathogenic oral microorganisms and the subsequent diagnosis of diseases, as well as the implementation of targeted therapeutic interventions. 

The objective of this review is to elucidate the most recent advancements in this domain, encompassing microbial colonization, state-of-the-art detection methodologies, and the intricate interplay observed during the progression of diseases. In the forthcoming decade, there is a promising potential for the efficient identification and targeted management of deleterious oral microorganisms.

## 2. Evolution: Microorganism to Microbiome

### 2.1. Evolution: Microorganism to Microbiome

The first recorded evidence of oral bacteria is credited to Antony van Leeuwenhoek, who, in 1676, offered detailed descriptions of microscopic organisms, referred to as “animalcules”, that were discovered in dental plaque samples. He noticed significant variations in the microbial makeup within the described habitats, as well as across samples collected from individuals displaying different degrees of health and illness, within these environments [14]. The term “microorganism” was thus coined to encompass various types of minutes living entities, including archaea, bacteria, protists, fungi, and viruses [15]. On the other hand, the “microbiome” is rather a collective term that serves as a comprehensive and inclusive designation for the entirety of microorganisms that exist within a certain environment [16]. These communities of microbes play a crucial role in maintaining human homeostasis and in disease development and progression. 

#### Evolution

Throughout the course of human evolution, the composition of our microbiome has been consistently influenced by our surrounding environment. Empirical evidence also shows that these resident microbes have been actively involved in metabolic processes within mammalian species for a period of no less than 500 million years. Coevolution in humans has also led to subtle yet significant variations among ethnic groupings [17,18].

The microbial populations present in the human oral cavity are influenced by a variety of biochemical and social variables. These influences encompass dietary habits, the environment, hygiene practices, physiological characteristics, medical status, genetic makeup, and lifestyle choices [19]. By tracing the evolutionary history of oral microbiota and exploring the underlying mechanisms, valuable insights can be gained to effectively address and manage diseases in contemporary times. 

For instance, the introduction of refined sugar into our diet during the early stages of agricultural development prompted specific oral bacteria to undergo genetic adaptations in their metabolic processes in response to the dietary shifts associated with the post-agricultural era [20]. An illustrative instance involves Streptococcus mutans, which exhibited the capacity to effectively outcompete other bacterial species in the oral cavity. This was achieved by the acquisition of defensive mechanisms against heightened oxidative stress and enhanced resistance to the acidic by-products resulting from its proficient glucose metabolism [20,21]. There is evidence suggesting that the composition of saliva and plaque in persons with caries exhibits more similarity compared to those who are healthy or have other oral diseases [22]. Furthermore, research also suggests that the oral microbiota, specifically some species within it, may have associations with systemic disorders, including Parkinson’s disease [23], type II diabetes mellitus [24], GI malignancies [25], inflammatory bowel disease (IBD) [26], and obesity [27]. 

Although initial research failed to establish a connection between human genetics and salivary oral microbiota, more recent studies have challenged this notion by investigating oral bacteria in dental plaque, analyzing discordant twin pairs, and exploring its link with specific human genomic alterations [28,29,30]. Collectively, these findings suggest a correlation between human genetics and the composition of microbiota, as well as the existence of certain species [29,30].

### 2.2. Oral Microbiome Composition

Humans, like other multicellular eukaryotes, are biological entities that encompass a multitude of microbial symbionts and their respective genomes [31]. The microorganisms residing within and on our bodies collectively constitute an integral organ that plays a crucial role in maintaining our overall well-being and physiological processes. In conjunction with our symbiotic microbial inhabitants, humans represent what is referred to as a “superorganism” or holobiont [1,31]. 

It is currently understood that the microorganisms, referred to as the microbiota, comprising the human microbiome exist not as coexisting individual unicellular organisms, but rather as intricately organized communities adhering to surfaces in the form of biofilms [1]. These biofilms exhibit a high level of regulation, both structurally and functionally, and are characterized by interactions among different species, including collaborations and antagonisms, which collectively contribute to the stability of the ecological system [32]. Bacterial cells residing in a biofilm possess the ability to engage in intercellular communication through the production, detection, and response to small diffusible signal molecules. This phenomenon, known as quorum sensing, bestows advantages such as facilitating host colonization, promoting biofilm formation, enhancing defense against competing organisms, and enabling adaptation to environmental fluctuations [33,34].

The intrinsic microbial communities in the human body are integral to critical physiological, metabolic, and immune functions, including but not limited to [35,36,37,38,39]:The process of development and differentiation of the host epithelium and defense mechanisms.Immune system development and regulation.Fine-tuning between pro-inflammatory and anti-inflammatory mechanisms in response to inflammation and infection.Promoting colonization resistance to prevent invasion and proliferation by infectious agents.Breaking down of complex carbs by the colonic microbiota, making them easier to absorb and assimilate and significantly contributing to host nutrition.Supplying energy and precursor molecules to produce mucosal lipids, as well as promoting the proliferation of epithelial cells, thereby preserving the integrity of the gastrointestinal (GI) tract.The interaction and detoxification of harmful contaminants or toxins like heavy metals, pesticides, cyanotoxins, etc.The modulation of gastrointestinal homeostasis via microbiota–immune system interactions, the loss of which can cause metabolic diseases such as obesity and type II diabetes.Maintenance of the gut–brain axis—microbiota products may affect the brain by creating regulating hormones or neurotransmitters; altering the gastrointestinal tract, autonomic nervous system, or intestinal nervous system; or boosting the immune system.The prevention of foreign infections through competitive eradication and antimicrobial factor(s) production via the human microbiota acting as an anatomical barrier.

#### 2.2.1. Bacterial Members

The oral cavity has a wide range of roughly 1000 bacterial species, largely representing different phyla. The taxonomic classification system encompasses six primary phyla. Actinobacteria (including Actinobacillum, Cryptobacterium, Tropheryma, etc.), Bacteroidetes (such as Tannerella, Prevotella, etc.), Firmicutes (including Parvimonas, Anaerococcus, Filifactor, etc.), Proteobacteria, Synergistetes, and Spirochaetes represent a wide range of microorganisms with various characteristics [39,40]. Microbiologists find these groups highly intriguing owing to their distinct traits and significant contributions within diverse ecological niches. The prevailing bacterial phylogenetics of the human oral microbiota, as derived from the Human Oral Microbiome Database (HOMD), can be summarized as follows [39,40,41,42,43]:**Firmicutes**: *Streptococcus, Lactococcus, Enterococcus, Lactobacillus, Gemella, Staphylococcus.***Tenericutes**: Mollicutes [G-1], Mycoplasma.**Firmicutes**: *Eubacterium, Peptostreptococcaceae, Mogibacterium, Filifactor, Parvimonas, Finegoldia, Anaerococcus, Peptoniphilus, Pseudoramibacter, Lachnospiraceae (G-1,2,3,7,8), Catonella, Oribacterium, Peptococcus, Oribacterium, Clostridiales, Selenomonas, Mitsuokella, Veillonellaceae (G-1), Veillonella, Dialister, Megasphaera.***Actinobacteria**: *Actinomyces, Rothia, Microbacterium, Propionibacterium, Mycobacterium, Gardnerella, Corynebacterium, Bifidobacteriaceae, Slackia, Cryptobacterium, Eggerthella, Atopobium.***Fusobacteria**: *Fusobacterium, Fusobacteria [G-1], Sneathia, Leptotrichia.***Bacteroidetes**: *Prevotella, Bacteroidaceae, Tannerella, Porphyromonas, Flavobacteriales, Bergeyella, Capnocytophaga.***Proteobacteria**: *Neisseria, Kingella, Simonsiella, Neisseria, Achromobacter, Bordetella, Lautropia, Burkholderia, Ralstonia, Delftia, Variovorax, Leptothrix, Stenotrophomonas, Xanthomonas, Cardiobacterium, Pseudomonas, Acinetobacter, Moraxella, Enterobacter, Escherichia, Klebsiella, Yersinia, Haemophilus, Aggregatibacter, Caulobacter, Caulobacter, Campylobacter.***Spirochaetes**: *Treponema.***Chlamydiae**: *Chlamydophila.***Chloroflexi**: *Chloroflexi [G-1].***Synergistetes**: *Jonquetella, Pyramidobacter, Synergistes [G-3].***TM7**: *TM7 [G-1, 2,3,4,5].***SR1**: *SR1(G1-1).*Archaea.**Euryarchaeota**: *Methanobrevibacter oralis.*

##### Candidate Phyla Radiation and the Enigmatic World of Microbial Dark Matter

The Candidate Phyla Radiation (CPR) is a large group of bacterial lineages that lack pure isolate cultures and are primarily defined through genome-resolved metagenomics [44]. CPR organisms exhibit common characteristics such as possessing small genomes and physical sizes, containing archaeal-specific RuBisCO genes, lacking certain metabolic enzymes, containing self-splicing introns within the 16S rRNA gene, and occupying deep subsidiaries within the bacterial subtree of life [45].

This hitherto undiscovered CPR group of organisms showcases a remarkable diversity, encompassing over 35 distinct phyla, inhabiting a wide range of ecological niches. They also lack the genes responsible for encoding a CRISPR/Cas bacteriophage defense system [45,46].

It is hypothesized that CPR organisms exhibit traits indicative of obligate symbiotic relationships. The dearth of direct evidence of their hypothesized symbiotic existence, coupled with a limited understanding of their physiological attributes, host interactions, and potential impact on the microbial community, can be attributed to their resistance to in vitro cultivation [8].

Representatives of the Candidate Phyla Radiation are GN02, SR1, and TM7 [47]. Numerous scholarly articles have documented the presence of CPR as prevalent in the human oral microbiota, exhibiting a greater proportion of “Candidatus Saccharibacteria” (TM7) compared to other microbial communities inside the human body [48]. Various assemblages of the phylum have been identified inside dental calculus [48,49], sulci of all teeth [49], keratinized and attached gingiva [50], the dorsal and coated regions of the human tongue [51], teeth surfaces (both healthy and with caries) [52], palatine tonsils, hard palate, the throat, and buccal mucosa, as well as in mouth rinse samples [53,54]. Currently, “Candidatus Saccharibacteria” is one of the most prevalent phyla found in the oral cavity of healthy individuals in contemporary human populations, accounting for roughly 3.1% ± 5.7% of the microbial diversity observed in saliva samples and 0.6% ± 1.2% of the bacterial population in dental plaque samples within a cohort of 200 persons who were deemed healthy [55].

The TM7 bacterium, originating from the human oral cavity, has been classified into six distinct groups, namely G1 to G6 [40,56]. The location and relative abundance of these groups within the microbiota exhibit variability. However, other phyla belonging to CPR, such as the phylum “Candidatus Absconditabacteria” (SR1), have also been identified. Even though the inclusion of this phylum in the core oral microbiota remains debated, numerous studies utilizing reverse genomics methods, 16S rRNA amplicon sequencing, and quantitative PCR have demonstrated its presence in various oral sites, such as saliva, tooth biofilms and plaque, subgingival fluid, etc. [50,52,53,54,55,56].

It is widely thought that the constituents of oral CPR (comprising of the core oral microbial taxa) play a significant role in shaping the oral microbial ecology. This is achieved through their ability to control the structural arrangement and operational behavior of the oral microbiome. These factors have been shown to have correlations with oral diseases, such as periodontitis and halitosis [45,46,57].

*Candida albicans*, a commensal fungus in the oral cavity, becomes problematic under conditions fostering dysbiosis. When the host’s immune defenses are compromised or oral hygiene is inadequate, *C. albicans* can transition to a pathogenic state, causing oral candidiasis. In denture wearers, Candida biofilm formation on acrylic surfaces is common, leading to denture-associated candidiasis. Ill-fitting dentures create microenvironments conducive to Candida overgrowth, resulting in inflammation and mucosal lesions. Persistent candidal infections can extend beyond the oral cavity, impacting systemic health. Regular oral hygiene practices, proper denture care, and immune support are crucial for preventing Candida-related oral complications [8].

#### 2.2.2. Mycobiome/Fungal Species

The empirical evidence substantiates the fact that fungi constitute a mere 0.004% of the collective oral microbiota. Candida, regardless of being a typical constituent of the vaginal mycobiont [58], has primarily been investigated for its colonization of the oral cavity [59]. This is due to its capacity to impact the composition of bacterial communities in initial oral biofilms, specifically facilitating the proliferation of obligate anerobic bacteria [60,61].

Several studies conducted by Dupey et al., Nasidje et al., Ghannoum, and Schuster et al. have examined the mycobiome composition of the oral cavity. These studies have collectively identified the presence of various fungal species, including Fusarium, Geotrichum, Hemispora, Aspergillus, Hormoedendrum, Penicillium, and Malassezia [8,46,47]. Additional fungus species that have been detected in the oral mycobiome include Cryptococcus, Cladosprorium, Claviceps, Clavispora, Cystofilobasidium, Davidiella, Emericella, Hormonema, Alternaria, Avena, Phoma, Zygosaccharmoyce, Schizosaccharomyces, and others [62,63].

A recent investigation has revealed a wide range of non-culturable fungal species present in the oral mycobiome of persons who are in good health. These organisms encompass many taxonomic groups such as Ascomycete, Basidiomycete, Glomus, Glomeromycete, Leptosphaeriaceae, Ectomycorrhiza, and others [64,65].

#### 2.2.3. Virome

The viral composition of the human oral microbiome exhibits significant inter-individual diversity and demonstrates remarkable temporal stability [19,43]. The oral virome is composed of eukaryotic viruses and bacteriophages, as elucidated by Caselli et al. in 2020 [66]. Herpesviridae, Papillomaviridae, and Anelloviridae are prominent families of eukaryotic viruses that are frequently encountered in various host organisms [65,67]. These viral families have been observed to exhibit a state of asymptomatic infection in individuals who are in good health [68]. Conversely, the phages exhibit greater diversity and have predominantly been investigated due to their ability to induce bacterial lysis, rendering them valuable for therapeutic applications against bacterial infectious diseases [69].

#### 2.2.4. Oral Microbiome Databases

To gain a deeper comprehension of the role played by the oral microbiota in both oral and systemic diseases, the United States has established the Human Oral Microbiome Database (HOMD) [8,39]. The primary aim of this database is to provide the scientific community with a comprehensive collection of microorganisms specifically found in the various anatomical regions of the human oral cavity [8]. 

The database has been carefully compiled, consisting of 619 taxa belonging to 13 phyla which have been recognized using a rigorous provisional nomenclature scheme based on the 16S rRNA gene and then assigned a unique Human Oral Taxon (HOT) number [8,70]. The HOT architecture effectively incorporates several types of data, including evolutionary, genomic, phenotypic, therapeutic, and literary information, for each taxonomic entity [70]. A BLAST search engine is given to facilitate the alignment of user-supplied 16S rRNA gene sequences with a rigorously maintained and extensive archive of whole 16S rRNA gene reference information [71,72].

Given that the primary focus of the Human Oral Microbiome Database (HOMD) is on the oral microbiome of the United States population, it is crucial to acknowledge that its findings may not provide an accurate representation of the oral microbial composition in other populations [4]. This limitation arises from the inherent variances observed in individual microbiomes. Several nations are presently establishing population-based databases that focus on inherent population characteristics, such as the Oral Microbiome Bank of China (OMBC). The OMBC represents an initial comprehensive account of the microbiome linked with the Chinese population [73].

Comparisons of bacterial 16S rDNA sequences to GenBank and other big open-access databases are typically insufficient for species identification and taxonomic classification [73]. Studies of the oral microbiome, which comprises many taxa, require good sequence data identification. CORE, a phylogenetically curated 16S rDNA database of the core oral microbiome, was created for this. The goal was to incorporate a comprehensive and minimally redundant depiction of oral bacteria with computationally robust species and genus classification [71,74].

#### 2.2.5. Oral Microbiome: Relation to Oral and Systemic Diseases

The fundamental aspect of maintaining optimal health lies in a sustainably balanced and diversified microbiome, which engages in commensalistic interactions among its congener microorganisms and mutualistic associations with its host organism [74,75]. Commensalistic interactions among microorganisms enable them to thrive without imposing any burdens on their host organism, hence contributing to the preservation of biodiversity [76]. The initiation and progression of diseases caused by oral bacteria are generally facilitated by synergistic or cooperative processes with interspecies interactions within the oral community, playing a pivotal role in defining the pathogenicity of oral microbiota [77,78].

Significant changes have been uncovered in the microbial composition of dental plaque among individuals with periodontal diseases, revealing an increase in pathogenic bacteria such as *Porphyromonas gingivalis*, *Treponema denticola*, and *Tannerella forsythia*, coupled with a decline in beneficial bacteria. Additionally, a dysbiotic microbial community encourages the formation of complex biofilms on tooth surfaces and beneath the gumline, providing an ideal environment for the survival and proliferation of pathogenic bacteria, leading to heightened virulence and tissue damage. Furthermore, dysbiosis not only impacts the microbial community but also modulates the host immune response, triggering chronic inflammation and contributing to the destruction of periodontal tissues. Genetic and environmental factors have been identified as potential contributors to microbial dysbiosis, necessitating a nuanced understanding of the development of personalized strategies in preventing and managing periodontal conditions. The identification of specific microbial species associated with periodontal diseases has opened doors to targeted therapeutic interventions, with ongoing research exploring approaches like probiotics, prebiotics, and antimicrobial agents to restore a balanced oral microbial community and promote overall oral health.

Human papillomavirus (HPV) infection is pervasive, with salivary HPV infection emerging as a noteworthy concern in healthy individuals. Although typically associated with genital and oropharyngeal cancers, recent research underscores the prevalence of asymptomatic salivary HPV infection in individuals without apparent health issues. The oral cavity serves as a reservoir for various HPV types, and saliva acts as a potential vector for viral transmission. Healthy individuals can harbor salivary HPV infections, often transiently, due to factors like intimate contact or exposure to contaminated surfaces.

The significance of salivary HPV infection lies in its potential to contribute to oral and oropharyngeal malignancies. Persistent infection with high-risk HPV strains, such as HPV-16 and HPV-18, can lead to dysplastic changes in the oral mucosa, serving as a precursor to malignancy. Despite being asymptomatic in healthy carriers, the long-term implications underscore the importance of understanding salivary HPV dynamics. Risk factors, such as sexual behavior and a compromised immune status, may influence the persistence and progression of salivary HPV infections. Vigilance in monitoring and elucidating the factors governing salivary HPV infections in healthy individuals is crucial for preventive strategies and early intervention to mitigate the potential progression towards malignancies associated with persistent high-risk HPV infections.

The intricate connections between oral health and overall well-being become apparent as the imbalance in the oral microbiota is linked to the initiation and advancement of diverse health issues. This visual representation underscores the importance of maintaining a healthy oral microbiome as a crucial aspect of preventive healthcare (Figure 1).

### 2.3. Methodologies Employed in the Study of the Oral Microbiome

#### 2.3.1. Sampling

The oral cavity harbors an intricately complex microbiome, each organism occupying a particular niche that possesses specific nutrient and environmental conditions that are conducive to the growth and survival of microbial organisms [4,8,72].

Accordingly, it is imperative to employ appropriate equipment to sample each micro-habitat [79]. In the context of oral mucosa, studies have documented the utilization of sterile microbrushes made of nylon, sterile Gracey curettes (sampling dental plaque residing on the hard tissues of teeth), paper cones, toothpicks, and unwaxed floss [80,81,82]. In terms of saliva sampling, it is important to highlight that a considerable amount of research studies predominantly concentrate on the examination of unstimulated saliva [11,83]. Nevertheless, it is crucial to recognize that alternative research endeavors have chosen to employ oral rinse samples, which entail the collection of saliva after rinsing the oral cavity with water [84].

#### 2.3.2. Microbial Cultivation and Microscopic Examination

The cultivation of isolated colonies has long been regarded for numerous decades as a key element in research, in understanding the physiological and pathogenic capabilities demonstrated by specific microbial taxa, and in clinical utilization in the field of oral microbiology [85,86].

Current approaches in microbial cultivation involve the utilization of nutrient-scarce or nutrient-free culture media extensions of cultivation periods, the utilization of serial dilution techniques to isolate bacterial strains with slow growth rates, the inclusion of specific growth-promoting factors in the culture media, and the creation of in vivo incubation conditions [87,88].

In the past few years, notable progress has been made in culturing methodologies, which involve upgraded sterilization techniques aimed at mitigating the presence of growth-inhibiting factors, culture media precisely customized for oral microorganisms, etc. [89,90,91]. An additional noteworthy advancement pertains to the co-cultivation of bacteria that secrete metabolites in conjunction with previously unexplored species, which have now acquired the capacity to flourish as satellite organisms in the presence of these symbiotic microbial entities [92,93,94].

##### Fluorescence In Situ Hybridization (FISH)

Fluorescence in situ hybridization (FISH) is a widely employed cytochemical non-radiological technique utilized for genetic recognition, identification, and positioning through the utilization of a nucleic acid probe that has been labeled with a fluorescent marker. This technique finds utility in the context of hybridization operations involving nucleic acid sequences of interest [95,96,97].

Accurate detection and analysis are facilitated through the utilization of fluorophores that exhibit closely overlapping excitation and emission spectra. The combinatorial labeling approach entails the utilization of a preset repertoire of several fluorophores to selectively label a particular bacterium [98,99,100,101]. By implementing this combinatorial labeling technique, the range of distinguishable microbial taxa within a specific visual area is substantially increased [99,100].

It is crucial to recognize the ongoing importance and necessity of bacterial culture in the field of microbiology, as a considerable fraction (31%) of the currently acknowledged oral taxa cannot be cultivated in laboratory settings [88,93]. Despite this, when it comes to diagnostic applications, while they are susceptible to antibiotics, culture-dependent methods are found to be time-consuming, costly, and less comprehensive in comparison to molecular DNA-based technologies. These molecular methods eliminate the need for cultivation [93,94].

### 2.4. Molecular Oral-Microbiology-Culture-Independent Approaches

The revelation of the double-helical arrangement of the DNA molecule, a groundbreaking discovery made by regarded Nobel laureates James Watson and Francis Crick in 1953, stands as an indisputably significant scientific revelation of the 20th century [102]. This groundbreaking investigation established the groundwork for individuals seeking to discern, classify, characterize, and comprehend microorganisms to advance culture-independent methodologies and nucleic-acid-centric molecular technologies [102,103]. As a result, the incorporation of molecular methodologies in this field has greatly enhanced our comprehension of the extensive oral microbial diversity [104]. This scientific breakthrough has brought to light the presence of complete bacterial phyla that have evaded isolation in a state of pure culture [103,104].

#### 2.4.1. DNA–DNA Hybridization

DNA–DNA hybridization (DDH) tests have been conducted since the 1960s to assess the extent of association among bacteria, widely due to its universal applicability and ability to provide comprehensive genome-wide comparisons between species [105,106]. The basis of these methods is the inherent capability of probe nucleic acids to exhibit specific and selective binding with their corresponding target nucleic acids. Probe nucleic acids are a class of nucleic acids, either DNA or RNA, that exist as single strands or as oligonucleotides and possess recognized sequences [106,107,108]. The fundamental principle posits that there exists a direct correlation between the level of gene expression and the quantity of labeled targets, thereby resulting in an amplified output signal [106,107]. Most contemporary techniques for nucleic acid detection employ labels that are linked to specific probes. The labels utilized in this context encompass fluorescent, chemiluminescent, or other functionalized modified molecules that possess the ability to emit optical signals, thereby serving as indicators of the hybridization event. Liposomes, magnetic beads, and gold particles have been employed as labeling agents in several studies [107,108,109].

Despite the unique advantages offered by DNA hybridization assays, the current DNA sensing methods that rely on these assays face several challenging issues. These include the consumption of large amounts of reagents, the requirement for lengthy and laborious procedures, and the reliance on bulky or expensive equipment [109,110].

##### DNA Microarray

The fundamental principle underlying microarray technology lies in the capacity for tagging nucleic acid molecules and subsequently employing them for the examination of other nucleic acids affixed to a solid support [109].

In conventional methodologies, wherein radioactive labeling agents are typically employed, the concurrent hybridization of test and reference samples is unattainable [108,109,110]. Recent technological advancements have facilitated the downsizing of probe detection methodologies for DNA, enabling the detection of numerous DNA or RNA sequences in a single experimental setup. The procedure exhibits an inverse nature compared to Southern blotting, wherein the probe is meticulously positioned upon a stationary substrate and subsequently subjected to the unbound nucleic acid, referred to as the target, for meticulous analysis [109,111] [Table 1].

#### 2.4.2. Polymerase Chain Reaction (PCR) Amplification

The polymerase chain reaction (PCR) is an innovative technique used for in vitro replication, enabling the rapid amplification of targeted genetic sequences [112]. This process leads to the production of millions of identical copies within a very brief timeframe of 2–3 h [112,113]. The utilization of the polymerase chain reaction (PCR) has been widespread in the field of diagnostic microbiology due to its notable sensitivity and specificity, together with its user-friendly characteristics and efficient processing time. A variety of PCR approaches have been developed to address different aims, such as nested PCR, asymmetric PCR, qualitative PCR, and reverse-transcription PCR [113,114,115] [Table 2].

### 2.5. Next-Generation Sequencing

NGS is a versatile, essential, and ubiquitous biological instrument that has influenced many biological fields. Next-generation sequencing (NGS), also known as high-throughput sequencing, is a powerful and revolutionary technology used in molecular biology to determine the precise sequence of nucleotides (A, T, C, and G) in DNA and RNA molecules [117,118,119,120,121,122].

Unlike traditional Sanger sequencing, which reads DNA sequences one strand at a time, NGS technologies can simultaneously sequence millions of DNA fragments in a single run, making it much faster and cost-effective [122,123].

The principal advantage of next-generation sequencing (NGS) resides in its capacity to replace conventional techniques of pathogen characterization. NGS employs a genomic-oriented approach to define pathogens [124]. The genomic compositions of pathogens delineate their identity, potentially encompassing valuable insights into their susceptibility to therapeutic agents and elucidating the interconnections among diverse pathogens. Such knowledge can be harnessed to track the origins of infectious outbreaks [123,125] and allows for the identification of disease-causing mutations and personalized medicine. It is used in cancer genomics, prenatal testing, and rare disease diagnosis [126]. Nonetheless, NGS poses several obstacles, including the production of substantial volumes of data, which necessitates computationally costly processes for data management and analysis. Furthermore, it is important to consider potential challenges related to the accuracy of reading, the comprehensiveness of coverage, and the presence of biases in the sequencing of information [126].

Thus, NGS has revolutionized the field of genomics, enabling large-scale studies and discoveries that were previously impractical or cost-prohibitive. It has led to a deeper understanding of genetic variations, gene expression, and the molecular mechanisms underlying various diseases and biological processes [127].

Several NGS platforms are available, such as Illumina, Pacific Biosciences (PacBio), Oxford Nanopore, and others. Each platform has its advantages and limitations, leading to diverse use cases [123,126,127,128,129] [Table 3].

### 2.6. Amplicon Sequencing as a Method for Systematic Characterization of the Microbiome

Amplicon sequencing, a DNA sequencing technique with high-throughput capabilities, has become a valuable tool for the comprehensive analysis of the microbiome. Genomic DNA fragments are subjected to massively parallel sequencing, yielding a substantial amount of sequence data on the order of gigabases [130].

The data obtained exhibit an exceptional level of sequencing depth when compared to the usual method of cloning and sequencing. Bacterial community profiling involves the utilization of hypervariable areas within the 16S rRNA, a minor portion of the ribosomal gene, which are then employed in the development of primers to target a broad range of bacterial populations (often referred to as universal bacterial primers) [130,131]. The hypervariable regions’ sequences are then utilized for distinguishing between several bacterial species. The sequence data obtained from the experiment are required to be processed via a bioinformatics pipeline [131,132,133].

The primary objective of this pipeline is to implement a process that effectively eliminates low-quality sequences and generates meaningful groupings or clusters of sequences, commonly referred to as operational taxonomic units (OTUs) [132]. The representative sequence of each operational taxonomic unit (OTU) is subsequently cross-referenced with sequences included in publicly accessible databases, such as the Ribosomal Database Project (RDP), and a taxonomic lineage (such as genus, family, or higher taxon) is assigned to the OTU [132,133].

### 2.7. Long-Read Sequencing

Long-read sequencing technologies are a class of sequencing methods that enable the generation of DNA sequences with longer read lengths compared to traditional short-read sequencing technologies and can produce uninterrupted sequences directly from native DNA, spanning from ten kilobases to several megabases in length [134].

In recent years, there has been a significant emergence of long-read, single-molecule DNA sequencing technologies, which have shown to be highly influential in the field of genomics [135]. The platforms under consideration have demonstrated their capability to accurately generate long reads ranging from tens to thousands of kilobases. This has enabled the resolution of complex regions within the human genome, the detection of structural variants that were previously unattainable, and the production of comprehensive assemblies of entire chromosomes from telomeres to telomeres [136].

Recent advancements in throughput and accuracy have significantly enhanced the usefulness and practicality of these technologies. Shortly, the utilization of long-read sequencing technology is expected to enable the regular construction of diploid genomes [136,137]. This advancement holds the potential to significantly transform the field of genomics by providing comprehensive insights into the complete range of human genetic variations. Consequently, it will address certain gaps in our understanding of heredity and facilitate the identification of previously unknown disease processes [136].

PacBio and ONT sequencing technologies can generate reads that can effectively navigate through the highly repetitive portions of the human genome. However, variations in their chemical composition and methods of sequence detection can impact factors such as read lengths, base accuracies, and throughput [138,139,140,141,142].

There is a need to devise innovative and sustainable strategies for managing the oral microbiome, focusing on the antimicrobial efficacy of various treatments and the promotion of oral health through green dentistry and natural polymers.

Silicone dental impression materials are prone to contamination by various microorganisms. Traditional disinfection methods often involve the use of chemical disinfectants, which can have environmental and health impacts. Recent studies have explored alternative disinfection methods, such as UVC radiation and gaseous ozone, for their ability to effectively eliminate pathogens without the drawbacks of chemical use. UVC radiation has shown promise in its ability to disrupt the DNA of microorganisms, leading to their inactivation, while gaseous ozone offers a potent oxidizing action that destroys microbes. Comparatively, these methods are effective in reducing the microbial load on dental impression materials, offering safer and more environmentally friendly alternatives to liquid chemicals [143].

The shift towards green dentistry emphasizes the use of sustainable and biocompatible materials, including the development of organic toothpaste formulations. A literature review on organic toothpaste highlights the benefits of using natural ingredients, such as plant extracts and essential oils, which possess inherent antimicrobial properties. These natural formulations not only contribute to oral health by reducing the microbial load but also have a lower environmental impact compared to traditional toothpastes that contain synthetic chemicals. This review underscores the importance of further research into the efficacy of these organic formulations in maintaining oral hygiene and health [144].

Natural polymers, such as chitosan, have garnered attention for their biocompatibility, biodegradability, and antimicrobial properties. Chitosan coatings have shown promise as antimicrobial and antifungal agents in the oral cavity. These coatings can be applied to dental devices or used in mouthwashes and toothpastes to provide a protective barrier against pathogens. Their ability to inhibit the growth of bacteria and fungi contributes to the maintenance of a healthy oral microbiome, highlighting the potential of natural polymers in oral health care [145].

Platelet-rich plasma (PRP) therapy, traditionally used for its regenerative properties in various medical fields, has also been explored for its antimicrobial effects in dentistry. PRP is rich in growth factors that can enhance wound healing and tissue regeneration. Recent studies have indicated that PRP may also possess antimicrobial properties that can be beneficial in treating oral infections. The application of PRP in dental procedures could potentially reduce the need for antibiotics, offering a natural and effective alternative for managing oral pathogens [146].

While the use of antibiotics remains a common approach to combating oral infections, concerns over antibiotic resistance and the impact on the oral microbiome have led to the exploration of targeted delivery systems. Metronidazole-loaded porous matrices represent an innovative solution, providing localized treatment for periodontitis. These matrices allow for the controlled release of metronidazole directly to the infected site, maximizing the therapeutic efficacy while minimizing systemic exposure and potential side effects. This targeted approach not only enhances treatment outcomes but also supports the preservation of the oral microbiome’s balance.

## 3. Future Perspectives

While significant progress has been made in understanding the oral microbiome and its potential impact on both oral and general health, there remain notable gaps in existing research that warrant critical examination and suggest avenues for future studies.

One prominent gap lies in the need for more comprehensive longitudinal studies that follow individuals over extended periods. Existing research often relies on cross-sectional data, providing snapshots rather than a dynamic understanding of how the oral microbiome evolves and its implications for health outcomes. Longitudinal studies could offer insights into the causal relationships between changes in the oral microbiome and the development of health conditions such as diabetes, stroke, cardiovascular diseases, infectious diseases, and gut health.

Most of the current research tends to focus on the identification of microbial patterns associated with specific health conditions. While this is valuable, a more refined exploration of the functional dynamics and mechanisms underlying these associations is needed. Understanding how the oral microbiome actively contributes to the development or prevention of health issues is crucial for developing targeted interventions and therapeutic strategies.

There is also a gap in our understanding of the interplay between the oral microbiome and systemic health, particularly regarding the bidirectional relationship between oral health and conditions like diabetes and cardiovascular diseases. More research is needed to unravel the molecular and immunological mechanisms through which the oral microbiome influences these systemic conditions and vice versa.

In addition to systemic health, the link between the oral microbiome and gut health remains relatively unexplored. Given the emerging significance of the gut–brain axis and the intricate connections between the gut and oral microbiomes, investigating how the oral microbiome influences gut health and its implications for overall well-being is a promising avenue for future research.

To bridge these gaps, future studies should incorporate interdisciplinary approaches, combining microbiology, immunology, and molecular biology. Advanced technologies such as multi-omics analyses can provide a more holistic understanding of the complex interactions within the oral microbiome and its systemic effects. Moreover, collaborative efforts between oral health researchers and specialists in fields like cardiology, endocrinology, and gastroenterology can enhance the integration of oral health into broader healthcare paradigms.

Recent breakthroughs in molecular research have significantly advanced our understanding of the oral microbiome. By using advanced sequencing and analytical methods, scientists have deeply explored the molecular makeup of the oral microbiome. These advancements not only help us understand the microbial composition but also shed light on how the oral microbiome functions. Identifying key microbial patterns related to oral health and illness opens new possibilities for diagnosis and treatment. The use of multi-omics approaches has the potential to provide a comprehensive understanding of how the host and microbiome interact, revealing the delicate balance that maintains oral stability. These molecular approaches are expected to take oral microbiome research to new heights. Beyond dentistry, these technologies can impact various fields like immunology, microbiology, and personalized medicine. This foundation allows for focused interventions, personalized treatments, and innovative preventive measures to maintain oral health and reduce the risk of systemic disorders linked to oral imbalances.

## Figures and Tables

**Figure 1 microorganisms-12-00571-f001:**
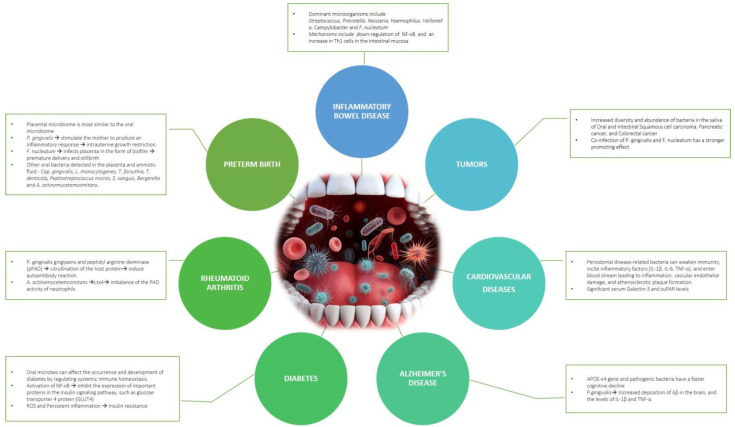
The figure depicts the impact of a dysbiotic oral microbiome on the initiation and advancement of various systemic diseases and conditions.

**Table 1 microorganisms-12-00571-t001:** Basic concepts of microarrays with their limitations and advantages.

Type	Properties	Limitations	Advantages
Oligonucleotide array Affymetrix GeneChips	Oligonucleotides are chemically synthesized on the array’s coated-quartz surface.	Limited to a few microbes—*S. cerevisiae*, *E. coli*, *B. subtilis*, *P. aeruginosa*, and *S. typhimurium*.	Cost-effective
	Enables very high feature densities—around 400,000	Others entail expensive custom designs by the company	The measurement of gene expression can be achieved by employing probes that cover the entire length of the transcript
Printed microarrays	Probes are produced individually and deposited onto the array by the utilization of a microarray spotter.	Expensive	Custom DNA microarrays allow for the creation of arrays for any species or strain
	Two distinct technologies are currently available: contact printers and noncontact printers	Spotters require a clean, controlled space with a regulated temperature and humidity—otherwise small liquid quantities evaporate quickly, making reproducible results difficult	It is feasible to generate variable quantities of arrays, modify slide chemistry, promptly adapt to advancements in annotation, eliminate probes for non-target genes, and incorporate probes that are specifically pertinent, such as those targeting intergenic areas
		High associated costs and special expertise needed—requires microarray core facilities to handle this task	
		Slight misalignments can cause variable or missing features.	
		Pins can only print a limited number of features before needing replacement	
		Cross-contamination and capillary blockage cause missing areas	
Double-stranded DNA microarrays	PCR amplification often produces double-stranded DNA.	Lengthy and laborious PCR product manufacturing and probe identity inaccuracies from generation faults	Cheaper cost, greater hybridization specificity, and sensitivity.
	Recommended amplified DNA length is 200–800 bp; however, bigger pieces up to 1.3 kb are also effective	Research suggests that 1 to 5% of commercial cDNA microarray probes may have incorrect identities.	Essential when the organism’s sequence is unavailable
Checkerboard DNA–DNA hybridization (CKB) Mini slot (Immunetics, Cambridge, MA, USA).	The approach employed is a robust non-polymerase chain reaction (PCR) method that relies on the concurrent hybridization of 40 digoxigenin (DIG)–labeled whole-genome DNA probes.	Nonspecific target binding	Rapid, sensitive, and relatively inexpensive.
	The process entails the extraction of DNA from oral samples, followed by the hybridization of the extracted sample with labeled probes that represent either the entire genomes or the 16S ribosomal RNA genes of bacteria with known identities.		

**Table 2 microorganisms-12-00571-t002:** Overview of different techniques of polymerase chain reactions.

Type	Principle	Application	Advantages	Limitations
Amplified fragment length polymorphism AFLP-PCR	Developed in 1995 by Peter Vos et al. [116]	To examine genetic diversity among species or closely related species, infer phylogenies at the population level, discern biogeographic patterns, create genetic maps, and determine the relatedness of cultivars	AFLP markers can analyze several loci simultaneously	Cannot detect poor DNA quality or degraded DNA.
	Combination of restriction-based and PCR-based methods		Organism sequence information is not necessary for designing primers complementary to adapter sequences	Cannot detect homozygous or heterozygous individuals due to its dominant marker nature.
	Uses restriction endonucleases to digest genomic DNA, followed by adapter ligation and PCR amplification		Requires a less genomic template	AFLPs are multi-locus, making it difficult to identify which fragment belongs to which DNA locus
			Highly reproducible outcomes with high-quality DNA input.	
Hot-start PCR	Variation of standard PCR that limits one reagent till heating to reduce non-specific binding	Inhibits hot-start Taq DNA polymerase activity or modified dNTP incorporation during reaction setup until heat activation	The reaction can be prepared at ambient temperature	The extended duration of heat exposure in comparison to conventional PCR necessitates the application of additional heat, resulting in increased vulnerability of the template DNA to potential damage
	Hot-start PCR avoids non-specific amplification and primer dimer formation. Increase results yield and accuracy		Enhanced productivity and accuracy	
	Magnesium-dependent			
Nested PCR	The utilization of nested PCR enhances the specificity of the reaction through the implementation of two distinct sets of primers, hence mitigating non-specific binding	Useful for pathogen detection and phylogenetic investigations; suitable for cancer and viral infection research	100% accuracy, specificity, and sensitivity	The process is time-consuming
			Beneficial for amplifying low-abundance genes.	Required more reagents such as an extra set of primer and one extra round of agarose gel electrophoresis. This makes the technique expensive
			Works well on impossible templates with high GC content or non-specific bands	
				Increased risk of contamination
Allele-specific PCR	Analyzes single-nucleotide polymorphisms using allele-specific primers	Detect the single-nucleotide polymorphisms (SNPs) at a particular location of the genome	Accurately distinguishes two alleles	Detects only known SNPs, not novel variations or mutations.
		Implements complicated primer design and mismatch incorporation		Higher false-positive rates necessitate regular internal, negative, and positive controls
		Accurately distinguishes homozygous and heterozygous alleles.		Includes extra primer sets, making the procedure expensive
		A crucial approach for genotyping and allelic variation research.		it cannot detect chromosomal alterations or bigger mutations such as deletions and duplications
	The ARMS-PCR, or allele-specific PCR, uses two primers for two alleles		Detects single-base variations (SNP)	
			The process is rapid, accurate, and reliable	
Multiplex PCR	Standard molecular biology technique for amplifying many targets in one test	Can be used to simultaneously amplify target sequences of different pathogenic microorganisms in a single reaction, with potential application in routine laboratories	Amplify many templates in one reaction or tube	High likelihood of non-specific binds and un-amplifications
	Uses a thermal cycler to amplify DNA using several primers and a temperature-mediated DNA polymerase		The technique is fast, efficient, and requires little labor	Has a substantial reaction failure risk if not executed effectively
			Multiplexing is inexpensive since it saves reagent, time, and power	Not every template (esp. long templates) can be multiplexed
			Each amplicon acts as an ‘internal control’ for another response, reducing false-positive results.	Primer self-inhibition
			Use less template material to provide more information	Low amplification efficiency
			Requires fewer consumables, chemicals, and utilities and additionally minimizes pipetting errors	Template efficiency differences. These factors would limit its development and use, especially in high-throughput GMO detection
Reverse transcription PCR (RT-PCR)	The RNA molecule is transformed into a complementary DNA (cDNA) molecule using the reverse transcriptase enzyme. This cDNA molecule is subsequently used as a template sequence in a polymerase chain reaction (PCR) reaction	RT-PCR is used mostly in gene expression research- and can be used in epigenetic, disease progression, and medication response investigations.	Sensitive because template RNA is amplified exponentially	Relative measurement of gene expression is limited by the need for a reference sample
		It can precisely measure disease-causing variations and estimate illness severity.	Gene-specific primers make RT-PCR cDNA synthesis very specific	Requires successful primer and probe design, a time-consuming and demanding procedure that demands standardization.
		Detect cancer biomarkers, severity, and progression	Yields fast results in one to two days	Alterations in reference and sample preparation affect gene expression accuracy
				Failure, primer-dimers, non-specific amplification, and imprecise quantification result from contamination
Quantitative PCR (qPCR) or real-time PCR	Quantitative polymerase chain reaction (Q-PCR) analyses integrate the conventional end-point detection PCR method with fluorescence detection technologies to monitor the buildup of amplicons in real-time throughout each cycle of PCR	Q-PCR assays can quantify ‘total’ bacterial (and/or archaeal) numbers by targeting highly conserved regions of the 16S rRNA gene while targeting taxa-specific sequences within hypervariable regions	Ability to quantify DNA quantities over a wide range, sensitivity, simultaneous sample processing, and immediate information	The machines cost more than typical PCR machines
			Highly sensitive, identifying even a single copy of the target nucleic acid sequence	Needs careful tuning of reaction parameters, such as primer design, annealing temperature, and enzyme concentration, which can be time-consuming
		qPCR is versatile, with applications in gene expression analysis, clinical diagnostics, pathogen identification, and food safety testing	High-throughput: qPCR enables examination of several samples simultaneously	qPCR may yield false-negative findings if the target sequence is absent or if inhibitors hinder the reaction
				Limited to DNA and RNA detection and quantification, not applicable to other biomolecules
Colony PCR	Colony PCR identifies in-plasmid DNA by generating primers	Identify proper ligation and insertion of DNA into bacteria and yeast plasmid	Rapid and affordable.	Any mutation or SNP within the ‘insert’ cannot be detected
		Gene transfer, treatment, cloning, genetic change, and CRISPR-CAS9-like investigations	The setup is straightforward, like PCR. No DNA extraction or plasmid purification is needed	It cannot give us sequence information
			More accurate and specific	The chances of false-positive results are very high
			Avoid tedious, time-consuming, and expensive restricted digestion.	
Digital PCR (dPCR)	Digital PCR (dPCR) is a sensitive and efficient method for measuring DNA or RNA levels in samples	Used in clinical specimens for determining the number of DNA and RNA viruses, bacteria, and parasites when well-calibrated standard is not available	Absolute measurement no standard curve	Limited dynamic range
			High sensitivity	High cost
			Improved PCR inhibitor resistance	
Repetitive element sequence-based PCR (rep-PCR)	Short repeated sequence regions throughout the bacterial genome are used to create oligonucleotide primers	DNA fingerprinting and bacterial strain classification using a general typing approach and genotype profile analysis	Fast, cheap, and specific; appropriate for anonymous genomic analysis	Fast, cheap, and specific; appropriate for unknown genomic analysis
			Profound segregation	Little discriminating power
			Inexpensive	Replicability may be lacking
Multi-locus sequence typing (MLST)	Comparison of test and reference strain PCR-amplified housekeeping gene sequences	Differentiating species strains	Unambiguity and transferability of sequence data	High cost of DNA sequencing.
			Scalability from a single bacterial isolate to many hundreds or even thousands of samples	

**Table 3 microorganisms-12-00571-t003:** Overview of different systems with their limitations and advantages.

System	Properties	Advantages	Limitations
Roche 454 System GS FLX Titanium system	The first commercially effective contemporary system.	Roche’s sequencing speed—10 h—is its biggest advantage.	Exorbitant expenses associated with reagent data
	The sequencer uses pyrosequencing which detects pyrophosphate produced during nucleotide incorporation to stop chain amplification instead of dideoxynucleotides.	Compared to other NGS systems, the read length is also unique	
AB SOLiD System (Sequencing by Oligo Ligation Detection)	The sequencer uses ligation-based two-base sequencing.	Exhibits the highest level of precision, without any dependence on a polymerase enzyme data	Implementation of this system necessitates substantial financial resources and the provision of an air-conditioned data center.
	Encompasses various areas of genomic study, such as epigenomics, whole-genome repeated sequencing, targeted resequencing, and transcriptomics. This includes investigations into gene expression profiling, small RNA analysis, and comprehensive analysis of the entire transcriptome.		Requires the utilization of a computing cluster consisting of four nodes, a team of proficient computing personnel, a distributed memory cluster, high-speed networks, and a batch queuing system
Illumina GA/HiSeq System	Uses synthesis sequencing—denatured single strands from the library with fixed adaptors are grafted to the flowcell and bridge-amplified to generate clonal DNA fragment clusters	Multiplexing in P5/P7 primers and adapters allows it to process thousands of samples.	Data
Ion PGM from Ion Torrent MiSeq—Illumina	Exhibits a small physical size and demonstrates rapid turnover rates, although it possesses a restricted capacity for data transmission	Intermediate yields can reach a maximum of 1 billion base pairs	The read lengths are rather modest, ranging from 200 to 400 base pairs (bps)
	These technologies are specifically designed for utilization in clinical settings and smaller laboratory environments	Possible to construct a maximum of five million sequences	The genome sequence is susceptible to the occurrence of base homopolymer runs, which has the potential to result in misassemblies
		Great accuracy—surpassing 99%.	
		Rapid execution durations (less than 8 h)	
Single-molecule real-time (SMRT) Pacific Biosciences	Ability to effectively sequence tiny genomes and analyze the closure of bacterial genomes without the need for further experiments	The utilization of faster and longer read durations facilitates the identification of nucleotide alterations	The cost of this item is quite high and it requires a significant amount of storage and computational resources
Single-molecule fluorescent sequencing Helicos	The single-molecule florescent sequencing (SMS) technique simplifies the process of preparing DNA samples and mitigates errors	Errors that arise as a result of the amplification process are eliminated	Utilization of this service is not widespread.

## Data Availability

No data available to share.

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
