# Peer review of "Microbial Symphony: Navigating the Intricacies of the Human Oral Microbiome and Its Impact on Health"

_microorganisms, 2024, doi:10.3390/microorganisms12030571_

Round 1
Reviewer 1 Report
Comments and Suggestions for Authors
With interest I’ve read the paper “Microbial Symphony: Navigating the Intricacies of the Human Oral Microbiome and its Impact on Health”. The authors presented a narrative review of the studies related to research on the human oral microbiome
The topic is of interest to a range of readers among the researchers in the field of oral microbiology. The paper is well-written and as thorough, as a chapter from the textbook. However, some comments should be addressed.
- The text is overwhelmed by the complicated words and phrases that should better be replaced for the sake of clarity and conciseness which are the primary characteristics of the quality scientific writing
- Please, make sure, that all abbreviations are explained when they are first met in the text (e.g., line 117 - abbreviation “GI” is used for the first time without explanation)
- Many sections of the manuscript are too general and lack the details related specifically to oral microbiome. Despite writing about microbiological methods in general (the information, which should be more o r less familiar to any doctor from the university lectures), it would be better to focus on advances or challenges of these diagnostic methods in relation to oral microbiome.
I think that the paper can be published after the aforementioned improvements are made.
Comments on the Quality of English LanguageThe text is overwhelmed by the complicated words and phrases that should better be replaced for the sake of clarity and conciseness which are the primary characteristics of the quality scientific writing. It is difficult to read and understand, especially for a non-native speaker.
Author Response
Reviewer 1
comments and Suggestions for Authors:
With interest I’ve read the paper “Microbial Symphony: Navigating the Intricacies of the Human Oral Microbiome and its Impact on Health”. The authors presented a narrative review of the studies related to research on the human oral microbiome.
The topic is of interest to a range of readers among the researchers in the field of oral microbiology. The paper is well-written and as thorough, as a chapter from the textbook. However, some comments should be addressed.
- The text is overwhelmed by complicated words and phrases that should better be replaced for the sake of clarity and conciseness which are the primary characteristics of the quality scientific writing
Thank you for the reviewer’s observations and we adjusted as per requirement.
- Please, make sure, that all abbreviations are explained when they are first met in the text (e.g., line 117 - abbreviation “GI” is used for the first time without explanation)
Thank you for the reviewer’s observations and we adjusted as per requirement.
- Many sections of the manuscript are too general and lack details related specifically to oral microbiome. Despite writing about microbiological methods in general (the information, that should be familiar to any doctor from the university lectures), it would be better to focus on the advances or challenges of these diagnostic methods about the oral microbiome.
Thank you for the reviewer’s observation and opinion.
I think that the paper can be published after the improvements are made.
Less...
Comments on the Quality of English Language
The text is overwhelmed by the complicated words and phrases that should better be replaced for the sake of clarity and conciseness which are the primary characteristics of quality scientific writing. It is difficult to read and understand, especially for a non-native speaker.
Thank you for the reviewer’s observation and opinion. We have tried to write in simple English so that the students, clinicians, and researchers can graph the context easily.
Reviewer 2 Report
Comments and Suggestions for Authors
The article addresses a fascinating topic; however, I suggest considering some improvements to enhance comprehension and effectiveness. The length of the article might pose challenges in absorbing the information, particularly with the presence of complex tables. I recommend that the authors focus on the primary objective: elucidating the true significance of the oral microbiome and how its alterations impact health.
Furthermore, it is noted that the article relies primarily on a narrative review, lacking a critical analysis of the mentioned studies. I would advise the authors to incorporate a more critical approach, highlighting gaps in existing research and proposing potential directions for future studies.
Conciseness is crucial; therefore, I suggest that the authors simplify the language when possible, avoiding unnecessary technical terms and ensuring the message is accessible to a broader audience. This will contribute to a more fluid and understandable reading experience.
In summary, the main focus should be on clear communication, incisive analyses, and an emphasis on the impact of changes in the oral microbiome on health. By incorporating these suggestions, the authors can significantly enhance the quality of the article.
Author Response
Reviewer 2
The article addresses a fascinating topic; however, I suggest considering some improvements to enhance comprehension and effectiveness. The length of the article might pose challenges in absorbing the information, particularly with the presence of complex tables. I recommend that the authors focus on the primary objective: elucidating the true significance of the oral microbiome and how its alterations impact health.
Thank you for the reviewers’ comments. We have tried editing the length of the document and also emphasized the true significance of oral microbiome and how its alterations would impact health in the following;
“The oral microbiome, a complex ecosystem of diverse microorganisms residing in the oral cavity, plays a pivotal role in maintaining oral health. Its intricate balance is crucial for immune modulation, nutrient metabolism, and prevention of pathogenic invasion. Alterations in the oral microbiome, often induced by factors such as diet, hygiene practices, or systemic diseases, can lead to dysbiosis, fostering the proliferation of harmful bacteria. This dysregulation is implicated in various oral health disorders, including periodontal diseases and dental caries. Beyond the oral cavity, emerging evidence suggests systemic implications, linking oral dysbiosis to conditions such as cardiovascular diseases and diabetes. Understanding and modulating the oral microbiome is paramount for holistic health maintenance”.
Furthermore, it is noted that the article relies primarily on a narrative review, lacking a critical analysis of the mentioned studies. I would advise the authors to incorporate a more critical approach, highlighting gaps in existing research and proposing potential directions for future studies.
Thank you for the reviewers’ comments. We have added the following as requested:
While significant progress has been made in understanding the oral microbiome and its potential impact on both oral and general health, there remain notable gaps in existing research that warrant critical examination and suggest avenues for future studies.
One prominent gap lies in the need for more comprehensive longitudinal studies that follow individuals over extended periods. Existing research often relies on cross-sectional data, providing snapshots rather than a dynamic understanding of how the oral microbiome evolves and its implications for health outcomes. Longitudinal studies could offer insights into the causal relationships between changes in the oral microbiome and the development of health conditions such as diabetes, stroke, cardiovascular diseases, infectious diseases, and gut health.
Most of the current research tends to focus on the identification of microbial patterns associated with specific health conditions. While this is valuable, a more refined exploration of the functional dynamics and mechanisms underlying these associations is needed. Understanding how the oral microbiome actively contributes to the development or prevention of health issues is crucial for developing targeted interventions and therapeutic strategies.
There is also a gap in our understanding of the interplay between the oral microbiome and systemic health, particularly regarding the bidirectional relationship between oral health and conditions like diabetes and cardiovascular diseases. More research is needed to unravel the molecular and immunological mechanisms through which the oral microbiome influences these systemic conditions and vice versa.
In addition to systemic health, the link between the oral microbiome and gut health remains relatively unexplored. Given the emerging significance of the gut-brain axis and the intricate connections between gut and oral microbiomes, investigating how the oral microbiome influences gut health and its implications for overall well-being is a promising avenue for future research.
To bridge these gaps, future studies should incorporate interdisciplinary approaches, combining microbiology, immunology, and molecular biology. Advanced technologies such as multi-omics analyses can provide a more holistic understanding of the complex interactions within the oral microbiome and its systemic effects. Moreover, collaborative efforts between oral health researchers and specialists in fields like cardiology, endocrinology, and gastroenterology can enhance the integration of oral health into broader healthcare paradigms.
Conciseness is crucial; therefore, I suggest that the authors simplify the language when possible, avoiding unnecessary technical terms and ensuring the message is accessible to a broader audience. This will contribute to a more fluid and understandable reading experience.
Thank you for the reviewers’ comments. We have revised it in as many places as possible.
In summary, the focus should be on clear communication, incisive analyses, and an emphasis on the impact of changes in the oral microbiome on health. By incorporating these suggestions, the authors can significantly enhance the quality of the article.
Thank you for the reviewers’ comments. We tried answering the queries put forth by the reviewers concisely and considerably.
Reviewer 3 Report
Comments and Suggestions for Authors
Dear Authors,
thank you for this interesting article. Here are some suggestions on how to improve it though:
1. When wrtiting about origins of oral microbiome, do not forget about the transmission between mother and child, see:
- Nardi GM, Grassi R, Ndokaj A, Antonioni M, Jedlinski M, Rumi G, Grocholewicz K, Dus-Ilnicka I, Grassi FR, Ottolenghi L, Mazur M. Maternal and Neonatal Oral Microbiome Developmental Patterns and Correlated Factors: A Systematic Review-Does the Apple Fall Close to the Tree? Int J Environ Res Public Health. 2021 May 23;18(11):5569. doi: 10.3390/ijerph18115569. PMID: 34071058;
Please, add that in the first two paragraphs of the study
2. Paragraph 2.1.1. is really interesting. I think those data occur not that frequently in the articles and I would like to congratulate you on the idea of introducing these informations.
3. Please, prepare graphic evaluation of HOMD - it would be more readable as diagram or table.
4. Pick better capitation name for 2.2.1.1
5. When writing about Candida, please refer to the most common C.albicans. Write more about why and when it could be problematic (eg. dentures)
6. In paragraph 2.2.5, write about HPV infection as the most common problem in healthy individuals - focus on Salivary HPV infection in healthy people
7. I would form a paragraph 3 (instead of conclusions) where I would like to see the perspectives and "looking up for the future"
8. I would like to see the chapter on "how to deal with oral microbiome" - the environment is really problematic, as there are plenty of different types of microorganisms, that could be pathogens to others. Please note the aspects of:
- Evaluation of Antimicrobial Efficacy of UVC Radiation, Gaseous Ozone, and Liquid Chemicals Used for Disinfection of Silicone Dental Impression Materials.
- Green dentistry: Organic toothpaste formulations. A literature review.
- Natural polymers in maintaining oral health (with chitosan coatings as one of the antimicrobiotic and antifungi speciments)
- Antimicrobial effect of PRP
- Use of antibiotics, among them loaded matrices,in particular Metronidazole-Loaded Porous Matrices for Local Periodontitis Treatment.
9. Add the limitations. The conclussions are formed correctly.
The article is full of details, written in the interesting way and I am sure it will find a lot of interests in the Readers, after adding the suggestions of Reviewers. Congratulations!
Author Response
Reviewer 3
- When writing about the origins of the oral microbiome, do not forget about the transmission between mother and child, see:
- Nardi GM, Grassi R, Ndokaj A, Antonioni M, Jedlinski M, Rumi G, Grocholewicz K, Dus-Ilnicka I, Grassi FR, Ottolenghi L, Mazur M. Maternal and Neonatal Oral Microbiome Developmental Patterns and Correlated Factors: A Systematic Review-Does the Apple Fall Close to the Tree? Int J Environ Res Public Health. 2021 May 23;18(11):5569. doi: 10.3390/ijerph18115569. PMID: 34071058;
Thank you for the reviewers’ comments. We have the following as requested.
The intricate process of oral microbiome transmission between mother and child begins during pregnancy and continues postnatally. Maternal factors, including mode of delivery, maternal health, and feeding practices, significantly influence the composition and development of the neonatal oral microbiome. Swallowing of amniotic fluid and early exposure to bacteria from the mother's oral and placental microbiomes shape the initial colonization of the fetal gut, subsequently influencing the establishment of the oral microbiome. Vertical transmission is evident through correlations between the maternal and neonatal oral microbiomes, emphasizing a direct link. Notably, this transmission is impacted by variables such as maternal exposure to disinfectants and antibiotics during delivery, type of delivery, maternal overweight status, and gestational diabetes. Although evidence is robust for certain factors, further exploration is required, particularly concerning maternal diet, and qualitative assessment of the maternal and neonatal oral microbiome dynamics.
- Paragraph 2.1.1. is interesting. I think those data occur not that frequently in the articles and I would like to congratulate you on the idea of introducing this information.
Thank you for the reviewer’s kind observations. We are thankful for the reviewer.
- Please, prepare a graphic evaluation of HOMD - it would be more readable as a diagram or table.
Thank you for the reviewer’s kind observations. We have inserted one figure illustrating the information required.
- Pick a better capitation name for 2.2.1.1
Thank you for the reviewer’s kind observations. We have inserted one appropriate caption for the figure illustrating: Candidate Phyla Radiation and the Enigmatic World of Microbial Dark Matter
- When writing about Candida, please refer to the most common C. albicans. Write more about why and when it could be problematic (eg. dentures)
Thank you for the reviewer’s observations. We have inserted the required information
Candida albicans, a commensal fungus in the oral cavity, becomes problematic under conditions fostering dysbiosis. When the host's immune defenses are compromised or oral hygiene is inadequate, C. albicans can transition to a pathogenic state, causing oral candidiasis. In denture wearers, Candida biofilm formation on acrylic surfaces is common, leading to denture-associated candidiasis. Ill-fitting dentures create microenvironments conducive to Candida overgrowth, resulting in inflammation and mucosal lesions. Persistent candidal infections can extend beyond the oral cavity, impacting systemic health. Regular oral hygiene practices, proper denture care, and immune support are crucial for preventing Candida-related oral complications.
- In paragraph 2.2.5, write about HPV infection as the most common problem in healthy individuals - focus on Salivary HPV infection in healthy people
Thank you for the reviewer’s observations. We have added the following
Human Papillomavirus (HPV) infection is pervasive, with salivary HPV infection emerging as a noteworthy concern in healthy individuals. Although typically associated with genital and oropharyngeal cancers, recent research underscores the prevalence of asymptomatic salivary HPV infection in individuals without apparent health issues. The oral cavity serves as a reservoir for various HPV types, and saliva acts as a potential vector for viral transmission. Healthy individuals can harbor salivary HPV infections, often transiently, due to factors like intimate contact or exposure to contaminated surfaces.
The significance of salivary HPV infection lies in its potential to contribute to oral and oropharyngeal malignancies. Persistent infection with high-risk HPV strains, such as HPV-16 and HPV-18, can lead to dysplastic changes in oral mucosa, serving as a precursor to malignancy. Despite being asymptomatic in healthy carriers, the long-term implications underscore the importance of understanding salivary HPV dynamics. Risk factors, such as sexual behavior and compromised immune status, may influence the persistence and progression of salivary HPV infections. Vigilance in monitoring and elucidating the factors governing salivary HPV infection in healthy individuals is crucial for preventive strategies and early intervention to mitigate the potential progression towards malignancies associated with persistent high-risk HPV infections.
- I would form a paragraph 3 (instead of conclusions) where I would like to see the perspectives and "looking up for the future"
Thank you for the reviewer’s observations and we adjusted as per requirement.
- I would like to see the chapter on "how to deal with oral microbiome" - the environment is problematic, as there are plenty of different types of microorganisms, that could be pathogens to others. Please note the aspects of:
- Evaluation of Antimicrobial Efficacy of UVC Radiation, Gaseous Ozone, and Liquid Chemicals Used for Disinfection of Silicone Dental Impression Materials.
- Green dentistry: Organic toothpaste formulations. A literature review.
- Natural polymers in maintaining oral health (with chitosan coatings as one of the antimicrobic and antifungal specimens)
- Antimicrobial effect of PRP
- Use of antibiotics, among them loaded matrices, in particular Metronidazole-Loaded Porous Matrices for Local Periodontitis Treatment.
Thank you for the reviewer’s observations.
- Add the limitations. The conclusions are formed correctly.
Thank you for the reviewer's comments and we have added the limitations section as requested.
The article is full of details, written interestingly and I am sure it will find a lot of interest in the Readers, after adding the suggestions of Reviewers. Congratulations!
Thank you for the kind reviewer’s observation and opinion.
Round 2
Reviewer 1 Report
Comments and Suggestions for Authors
Dear authors,
the manuscript has become better. Some details were added that made it even more interesting and comprehensive.
However, I see no attempt to simplify the text and wording. I believe that making your text concise and clear is still necessary.
Comments on the Quality of English LanguageEnglish is correct, but overwhelmed by difficult words. No improvement in this regard can be seen.
Reviewer 2 Report
Comments and Suggestions for Authors
The authors significantly enhanced the quality of the manuscript and incorporated many of the proposed suggestions. I am confident that the manuscript is now ready for submission for publication.
Reviewer 3 Report
Comments and Suggestions for Authors
Thank you for the corrections